# Randomised controlled trials for improving health outcomes for people living with multiple long-term conditions: Protocol for a systematic review of methodological approaches, risk of bias and reporting quality

Lisong Zhang[1,2], Elizabeth Fisher[1], Naomi Bradbury[1,2], Natalie Darko[2], Kamlesh Khunti[2,3], Selina Lock[4], Sally J. Singh[5], Sharon A. Simpson[6], Ellesha Smith[1], Susan M. Smith[7], Rod S. Taylor[6,8], Miles Witham[9], Hannah Young[10], Laura J. Gray[1,2]*

1 Department of Population Health Sciences, University of Leicester, Leicester, United Kingdom, 2 NIHR Leicester Biomedical Research Centre, University of Leicester, Leicester, United Kingdom, 3 Diabetes Research Centre, University of Leicester, Leicester, United Kingdom, 4 Library and Learning Services, University of Leicester, Leicester, United Kingdom, 5 Department of Respiratory Sciences, University of Leicester, Leicester, United Kingdom, 6 Medical Research Council and Scottish Chief Scientist Office, Social and Public Health Sciences Unit, School of Health and Well Being, University of Glasgow, Glasgow, United Kingdom, 7 Discipline of Public Health and Primary Care, Trinity College, Dublin, Ireland, 8 Robertson Centre for Biostatistics, School of Health and Well Being, University of Glasgow, Glasgow, United Kingdom, 9 Newcastle Biomedical Research Centre, Newcastle University, Newcastle, United Kingdom, 10 University Hospitals of Leicester, Leicester, United Kingdom

* lg48@le.ac.uk

## Abstract

### Introduction

The number of people living with multiple long-term conditions (MLTC or multimorbidity) is increasing. There have been national and international calls for more and better research in this clinical area. This systematic review will assess the methodological approaches, risk of bias and standards of reporting of the included trials. This study will benchmark current methodological practice and identify areas for improvement.

### Methods and analysis

MEDLINE, Scopus, CINAHL, CENTRAL and the International Research Community in Multimorbidity database will be searched from January 1999 to identify randomised controlled trials conducted with the aim of improving health outcomes for people living with MLTCs. Study screening, data extraction and the risk of bias assessment will be conducted independently by two reviewers. Data to be extracted will include study descriptors, design and analysis characteristics, methodological quality, bias and standard of reporting. A narrative synthesis will be conducted to summarise current methodological practice and to identify areas for improvement.

**Data availability statement:** NA protocol.

**Funding:** This study is funded by National Institute for Health and Care Research (NIHR) RfPB Under-represented disciplines and specialisms highlight notice: Methodologists: NIHR206813 – Identifying methodological uncertainties in the design and analysis of trials in people living with multiple long-term conditions. This study was supported by the National Institute for Health and Care Research (NIHR) Applied Research Collaboration East Midlands (ARC EM) and Leicester NIHR Biomedical Research Centre (BRC).

**Competing interests:** KK is National NIHR Applied Research Collaborations theme lead for Multiple Long Term Conditions and is Co-Chair of the NIHR Cross NIHR Collaboration for Multiple Long Term Conditions. SAS is President of the UK Society of Behavioural Medicine. She has also been a member of the NIHR HTA Clinical Evaluations and Trials Committee (2016-2020), the Commissioning Panel for the NIHR Policy Research Programme (2019-2022); the Chief Scientist Office Health Improvement Protection and Services committee (2018-2023). RST and SAS are currently co-chief investigators on an NIHR funded PERFORM programme (NIHR202020), developing and evaluating the impact of a exercise-rehabilitation intervention for people with multiple long-term conditions. LJG, KK and ND are Co-Is: Davies M (PI) Type 2 Diabetes (T2D) in 18 – 40-year-olds: A Multifactorial Management Intervention to Address Multimorbidity in Early-Onset T2D in Adults (The M3 Research Programme). NIHR Programme Grants for Applied Research. £2,570,619.00. Start date Aug 2021 (5 years).

## Ethics and dissemination

Ethical approval is not required. The results of the review will be published in a peer-reviewed journal and shared via conferences and webinars.

**PROSPERO registration number:** CRD42024595925

## Introduction

Multiple long-term conditions (MLTCs), also known as multimorbidity, are the co-existence of two or more long-term conditions (LTCs) [1–4]. While the definition of 'long-term' can vary, it generally refers to chronic conditions lasting at least 12 months [5,6]. MLTCs are an increasing public health concern, with 8.8 million people in England currently living with them [7]. By 2035, it is projected that two-thirds of adults aged 65 and older in England will be affected [8]. Although LTCs and MLTCs are often associated with aging, studies show that the age of onset has been decreasing over time [9]. MLTCs also contribute to the widening inequality gap, with a higher prevalence observed in ethnic minority groups [7,10] and deprived populations [7,9,11]. Rising life expectancy means that more people are living with MLTCs for longer periods [12]. People with MLTCs also experience a high treatment burden [13], poorer health-related quality of life (HRQoL) [14] and are more likely to die prematurely [15] compared to those who do not.

International and national research priorities, such as those set by the Academy of Medical Sciences and the National Institute of Health and Care Research (NIHR), include improving outcomes for people with MLTCs [1,16]. Despite the increasing number of people living with MLTCs, current care and research are primarily focussed on single conditions [17]. Given the complexity of MLTCs, there is an urgent need for novel approaches to gather evidence on the effectiveness of interventions. To date, relatively few trials have tested interventions aimed at improving outcomes for people with MLTCs, and those that have been conducted have not provided clear evidence to support widespread implementation [18,19].

People with MLTCs have specific needs beyond those who have single conditions [20]. An ethnographic study of individuals with various MLTCs identified common challenges, regardless of the number of co-occurring conditions. These challenges included mobility issues, the need for encouragement to take an active role in managing their conditions, and the importance of setting achievable goals [21]. The study also highlighted the significance of well-being and deprescribing. Through a priority-setting exercise involving people living with MLTCs, their carers, and healthcare professionals, the top 10 research priorities for people with multiple conditions in later life were identified. Four of these priorities focus on the need for information regarding the effectiveness and cost-effectiveness of interventions to reduce social isolation, prevent MLTCs, and improve exercise and psychological well-being [22]. Addressing these gaps and evaluating these complex interventions, will likely lead to more pragmatically designed trials in this area.

Although previous reviews have been conducted in MLTC trials, they have been focussed on intervention effectiveness both generically [18] and focussed on specific types of intervention (such as models of care [23], behaviour change interventions [24], medication adherence [25]), or on specific methodological issues (such as ethnic representation [26]) or on MLTC subgroups (such as those with frailty plus MLTC [27]). The aim of this research is to conduct a systematic review of all published randomised trials conducted in MLTC populations to identify methodological approaches to trial design and analysis as well as assessing the quality of their reporting and risk of bias.

## Materials and methods

This protocol is reported in line with the Preferred Reporting Items for Systematic Reviews – Protocols (PRISMA-P) guidelines (S1 File) and is registered on PROSPERO [28]. The methodology for this review has been informed by previous reviews conducted in this area [18,24,26,29,30]. Throughout this review MLTCs will be defined using the definition provided by Academy of Medical Sciences – The co-existence of two or more chronic conditions, including any combination of:

- A physical non-communicable disease of long duration, such as a cardiovascular disease or cancer.

- A mental health condition of long duration, such as a mood disorder or dementia.

- An infectious disease of long duration, such as HIV or hepatitis C [1].

The study is expected to commence in November 2024, with searching, selection and data extraction expected to be completed by July 2025. It is expected that the results from the review will be available by the end of 2025.

### Search

Searches will use the following electronic databases: MEDLINE (Ovid), Scopus, CINAHL (EBSCOHost) and CENTRAL (Cochrane Library). These databases will be searched from January 1999 to identify relevant trials. The start date of January 1999 was chosen, as this is when the first trial in MLTCs was reported in a previous systematic review [29]. The end date will be the date on which the searches are completed.

A copy of the searches can be viewed in S2 File. A validated filter to identify trials [31] will be used and combined with terms relating to multimorbidity and MLTCs, using the limit function to apply the date and English language restrictions. Terms related to comorbidity or the co-existence of long-term conditions will not be included in the search strategies, given that such studies will be excluded. Backward citation searches of the reference lists of the included studies and the previously published reviews will also be conducted [18,24,26,27,30]. The list of multimorbidity publications on the website of International Research Community in Multimorbidity will be hand searched [32].

### Inclusion criteria

Published randomised (e.g., individually or cluster randomised) controlled clinical trials of any design (parallel, cross-over, factorial etc) and in any setting with the following characteristics:

- Participants: Adults (aged 18 and over) with two or more long-term conditions. Only studies including adults are eligible for this review. MLTCs do occur in children, however the prevalence is much lower [33].

- Intervention: Any type of interventions that aim to improve the outcomes for people with MTLCs. Based on the previous reviews it is expected that most of the interventions will be complex in nature.

- Comparator: Treatment as usual or another novel intervention (i.e., head-to-head trials).

- Outcomes: Any health outcomes, including clinical events (such as mortality or hospitalisation) and patient reported outcomes (including but not limited to QoL, HRQoL and measurement of physical or mental functioning [34]).

Due to time and financial restraints, only studies published as full text articles in English will be included, the number of papers excluded due to language restraints will be reported. Studies published in English reporting trials conducted outside of the UK will be included.

## Exclusion criteria

Non-experimental studies such as observational studies, non-randomised studies (e.g., interrupted time series, non-controlled trials, before/after studies) and pilot or feasibility studies will be excluded. Early phase studies (1 or 2) will also be excluded. The focus of this review is on trials of MLTCs, therefore secondary analyses or subgroup analyses of previously published trials which focus on subsets of people with MLTCs will not be included. Cost effectiveness analyses of trials, meta-analysis, systematic reviews, interim analyses, as well as letters or commentaries will be excluded. Trial protocols and trials published as abstracts only will also be excluded.

In line with the reviews by Smith et al, trials which include a population with an index condition (i.e., studies for which the inclusion criteria is a named condition (diabetes for example) plus one or more other conditions) or focussing on comorbidity (i.e., studies for which the inclusion criteria is two named conditions (diabetes and hypertension for example)) will be excluded. Additionally, trials which focus on only the age of participants (i.e., older adults) without specifically addressing multimorbidity will also be excluded [18]. Studies which identify participants using the number of prescribed medications, or on the basis of their risk of using health services will be excluded, as will studies which focus on carers of people with MLTCs. No other restrictions relating to participant demographics or protected characteristics will be applied.

## Study selection

All eligible studies from the literature search will be imported into the online app Covidence [35] where duplicates will be removed. The titles and abstracts of each study will be screened independently by two reviewers.

Those articles progressing to the full text review, will be further screened, again by two independent reviewers. Those excluded at this stage will have the reason recorded. The screening and selection process will be captured in a PRISMA flowchart.

Trials published across two or more papers will be grouped together and counted as a single study. When possible, the main results paper will be used for data extraction, with other papers consulted to identify any missing information.

At all stages, disagreements will be resolved by a third reviewer.

## Data extraction

Data will be extracted in to a pre-piloted standardised data extraction form in Covidence. All data will be extracted independently by two reviewers, with disagreements discussed with a third.

The following data will be extracted from the included articles.

Study descriptors: Trial title, publication details (first author, year, journal), country, setting (community, primary care, secondary care), intervention(s) and control type, MLTC related eligibility criteria, number of participants/(clusters) recruited per arm, trial duration and follow up times, recruitment rate, retention rate, assessment of intervention compliance, summary statistics of participant characteristics (age, sex, ethnicity). The funder of the trial will be extracted. If the trial is of a cluster design, then details on the type and size of clusters will also be extracted.

Design characteristics: Trial design (parallel, cross-over, factorial etc), number of arms, purpose (effectiveness, non-inferiority, and equivalence), type of randomisation (individual, cluster), randomisation parameters (stratification, minimisation, blocking etc), achievement of allocation concealment, type of blinding, primary outcome, sample size factors

(n required, assumptions about MCID, SD, power, alpha, drop out and ICC if appropriate), use of existing data sources, number of data collection points and whether incentives were provided to participant, whether a feasibility/pilot study informed the design (internal or external). The rationale given for particular design choices and any reported patient and public involvement (PPI) in the design will be extracted as free text.

Analysis characteristics: Reporting of a pre-specified analysis plan, how LTCs were handled in the baseline table, approach to primary analysis (ITT, per protocol), approach to missing data, handling of intercurrent events, analysis method for primary outcome including whether any adjustment was undertaken, details of any sensitivity or subgroup analyses and whether these were prespecified, reporting of any analysis taking account of compliance to the intervention, adjustment for multiple testing, assessment of the appropriateness of the analysis conducted, assessment of trial design specific analysis requirements – such as accounting for clustering and reporting ICC for trials of a cluster design. Whether the authors consider the representativeness of the sample recruited for the trial and any methodological challenges that were highlighted by the authors will be extracted.

Risk of bias: The risk of bias of the included studies will be assessed using the Cochrane Risk of Bias (RoB) tool [36]. This tool assesses bias across five domains (selection, performance, attrition, reporting, other) and overall. Each domain is rated as either low, high or some concerns in terms of the risk of bias. For trials of a cluster or factorial design, the RoB tools for these specific designs will be used.

Standard of reporting: To assess the quality of reporting the CONSORT statement checklist will be completed for each study. Whether a completed CONSORT checklist was available for the paper will also be extracted. We will compare the data in the checklist completed by the study authors to the data extracted for this project.

## Data synthesis

A narrative analysis will be conducted to summarise the study design, analysis, reporting and bias characteristics of the included trials using summary statistics. Tables and figures will be used to illustrate and compare the main characteristics of the trials, the risk of bias and the quality of the reporting (CONSORT). Due to the nature of the research question, meta-analysis will not be conducted.

Subgroup comparisons to explore the heterogeneity by types of intervention, duration of intervention and types of MLTCs will be conducted.

The results will be interpreted taking the risk of bias and quality of reporting of the studies into account.

## Patient and public involvement

To inform our plans for this review, meetings with two groups were held online: (i) 11 public contributors all of whom live with MLTCs, (ii) 4 public contributors from our PPI-SMART (Patient and Public Involvement with Statistical Methods and Research Techniques) group, some of whom live with MLTCs. Both groups recognised the importance of the work and suggested extracting additional data. This included: missing data, drop out, duration of study, number of study interactions, use of incentives, whether online data collection was used, and whether studies reported PPI input. There were concerns about the representativeness of MLTC studies, the groups reported that having MLTCs could affect people's ability to contribute to research, particularly younger people with MLTCs who may have children, caring responsibilities or work. Whether this has been considered in the included studies will be assessed.

An independent steering committee has been established for this programme of work, which includes a public contributor.

## Ethics and dissemination

Ethical approval is not required for this review as it will use published existing aggregate data only. The results of the review will be published in a peer reviewed journal. Once the full programme of research is complete (review and following

consensus study), webinars to present the results and to encourage researchers to address the research uncertainties found will be held. In addition, the results will be shared via social media (LinkedIn and Bluesky).

## Results and discussion

This comprehensive review of the methodological approaches used in trials aimed at informing the care of individuals living with MLTCs responds to global calls for more research in this important and rapidly growing clinical area.

Currently, there is limited research to guide the design of studies in this complex and evolving area. In 2018, the Academy of Medical Sciences published its first international policy report on MLTCs, highlighting multimorbidity as a growing global concern but with limited evidence on effective interventions [1]. The UK cross-funder multimorbidity framework was published in 2020; they included facilitating MLTC inclusive clinical trials as a priority [37]. Trial advancement was also a feature of the NIHR strategic framework for MLTCs, with methodologies being identified as one of the strategic priority areas [16]. The NIHR have specifically noted the need for outcome frameworks and ensuring representation within MLTC trials. This review aims to benchmark current methodological approaches in MLTC trials and identify opportunities for improvement.

Unlike the previous reviews in this area, terms related to comorbidity will not be included in our searches. This was a pragmatic decision to limit the number of studies needed to be screened and given that our interest is in trials of interventions aimed at those with MLTCs in the broadest sense and not those living with an index condition plus another (i.e., comorbidity). Any studies which have used terms related to comorbidity to describe a MLTC population will be picked up through screening the references of previous systematic reviews in this area. As the terms 'multimorbidity' and 'MLTC' are becoming more widely adopted to describe this patient group, we anticipate this approach will have minimal impact on the inclusion of recent studies.

There are several methodological limitations that should be considered when interpreting the results of this review. Grey literature will not be searched, and articles not published in English will be excluded. These decisions were made due to resource constraints, which may result in the omission of relevant trials. However, since the primary goal of this review is to synthesise current practices in the design and analysis of trials in MLTCs, rather than estimate the effectiveness of interventions, this limitation is considered acceptable.

## Conclusions

This review is the first step in a research programme aimed at improving the design and analysis of trials in people with MLTCs. It will provide data for the first round of a consensus study to establish and prioritise the methodological uncertainties in this field. Once established, the priority-setting exercise is expected to stimulate more relevant and necessary methodological research, ultimately enhancing both the quality and quantity of studies focused on improving outcomes for individuals living with MLTCs.

### Strengths and limitations of this study

- A comprehensive review of all randomised trials testing interventions to improve health outcomes for people living with multiple long-term conditions will be conducted to assess their methodological approaches.

- Trials will be included irrespective of their setting.

- The protocol is reported according to the Preferred Reporting Items for Systematic Reviews and Meta-analyses Protocol guideline.

- The study screening, selection, data extraction and assessment of the risk of bias will be completed by two independent reviewers.

- Only English language publications of published randomised controlled trials will be included.

## Supporting information

**S1 File. PRISMA-P Checklist.**
(DOCX)

**S2 File. Search strategies.**
(DOCX)

## Author contributions

**Conceptualization:** Lisong Zhang, Darko Natalie, Kamlesh Khunti, Sally J Singh, Ellesha Smith, Laura Jayne Gray.

**Data curation:** Naomi Bradbury.

**Methodology:** Naomi Bradbury, Ellesha Smith.

**Writing – original draft:** Lisong Zhang, Elizabeth Fisher, Selina Lock, Sally J Singh, Laura Jayne Gray.

**Writing – review & editing:** Naomi Bradbury, Susan M Smith, Sharon A Simpson, Ellesha Smith, Rod S Taylor, Miles Witham, Hannah Young, Laura Jayne Gray.

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
