## [Decision Letter · Decision Letter 0]

26 Dec 2024

PONE-D-24-47106Randomised controlled trials for improving health outcomes for people living with multiple long-term conditions: protocol for a systematic review of methodological approaches, risk of bias and reporting qualityPLOS ONE

Dear Dr. Gray,

Thank you for submitting your manuscript to PLOS ONE. After careful consideration, we feel that it has merit but does not fully meet PLOS ONE’s publication criteria as it currently stands. Therefore, we invite you to submit a revised version of the manuscript that addresses the points raised during the review process.

We look forward to receiving your revised manuscript.

Kind regards,

Paolo Landa, Ph.D.

Academic Editor

PLOS ONE

Journal Requirements:

2. Please note that funding information should not appear in the Acknowledgments section or other areas of your manuscript. We will only publish funding information present in the Funding Statement section of the online submission form. Please remove any funding-related text from the manuscript. 

“KK is National NIHR Applied Research Collaborations theme lead for Multiple Long Term Conditions and is Co-Chair of the NIHR Cross NIHR Collaboration for Multiple Long Term Conditions.

SAS is President of the UK Society of Behavioural Medicine. She has also been a member of the NIHR HTA Clinical Evaluations and Trials Committee (2016-2020), the Commissioning Panel for the NIHR Policy Research Programme (2019-2022); the Chief Scientist Office Health Improvement Protection and Services committee (2018-2023).

RST and SAS are currently co-chief investigators on an NIHR funded PERFORM programme (NIHR202020), developing and evaluating the impact of a exercise-rehabilitation intervention for people with multiple long-term conditions. 

LJG, KK and ND are Co-Is: Davies M (PI) Type 2 Diabetes (T2D) in 18 – 40-year-olds: A Multifactorial Management Intervention to Address Multimorbidity in Early-Onset T2D in Adults (The M3 Research Programme). NIHR Programme Grants for Applied Research. £2,570,619.00. Start date Aug 2021 (5 years).”

4. We notice that your supplementary materials are included in the manuscript file. Please remove them and upload them with the file type 'Supporting Information'. Please ensure that each Supporting Information file has a legend listed in the manuscript after the references list.

**Additional Editor Comments:**

Many updates are required for improvement. Please follow the indications of the reviewer

Reviewers' comments:

Reviewer's Responses to Questions

Comments to the Author

1. Does the manuscript provide a valid rationale for the proposed study, with clearly identified and justified research questions?

Reviewer #1: Partly

2. Is the protocol technically sound and planned in a manner that will lead to a meaningful outcome and allow testing the stated hypotheses?

Reviewer #1: Partly

3. Is the methodology feasible and described in sufficient detail to allow the work to be replicable?

Reviewer #1: Yes

4. Have the authors described where all data underlying the findings will be made available when the study is complete?

Reviewer #1: Yes

5. Is the manuscript presented in an intelligible fashion and written in standard English?

Reviewer #1: Yes

6. Review Comments to the Author

You may also provide optional suggestions and comments to authors that they might find helpful in planning their study.

Reviewer #1: Thank you for the opportunity to review this paper. I think this paper is worth publishing following some updates, especially a major revision of the methods section.

7. PLOS authors have the option to publish the peer review history of their article (what does this mean? ). If published, this will include your full peer review and any attached files.

Do you want your identity to be public for this peer review? For information about this choice, including consent withdrawal, please see our Privacy Policy .

Reviewer #1: No

---

## [Author Response · Author response to Decision Letter 1]

19 Feb 2025

Response to reviewer’s comments

We thank the reviewer for their comments and insight, we have taken onboard these suggestions and updated the manuscript. Below we have addressed each comment in turn. Where required we have also made the methodology more explicit.

General comments

Comment: Please remove all references to “we” in the text. For example, on page 4, “Strengths and limitations”, “we will conduct A comprehensive review of all randomised trials testing interventions to improve health outcomes for people living with multiple long-term conditions to assess their methodological approaches was conducted”.

Response: All references to ‘we’ have been removed from the manuscript.

Comment: Cross-check the paper regarding full stops, grammar and general sentence structure and flow.

Response: The grammar and sentence structure and flow has been thoroughly reviewed.

Introduction

Comment: Page 5, line 76 “National research priorities include improving outcomes for people with MLTCs (1, 14). Despite more people living with MLTCs than single conditions, current care and research are largely single condition focused (15)” Please clarify whose national research priorities you’re referring to. Is this also a global issue?

Response: this has been clarified.

Comment: Page 5, line 83 “People with MLTCs have specific needs beyond those who have single conditions.” Consider adding a reference to support this.

Response: A reference has been added.

Comment: Page 5, line 88. “Of the top 10 James Lind Alliance….” Suggest you explain what this group is and how this is relevant.

Response: This has been clarified.

Methods and Analysis

Comment: Page 7, line 109, “Based on the conditions identified by Ho et al. (25). The study is expected to commence in November 2024, with searching, selection and data extraction expected to be completed by February 2025. It is expected that the results from the review will be available by the end of 2025.” It was not clear what the conditions identified by Ho are; please clarify or reword these sentences.

Response: We have removed this sentence as we agreed it is not clear.

Comment: Page 7, line 114 “We will search the following electronic databases: Ovid MEDLINE, Scopus, CINAHL and Cochrane Library from January 1999 to identify relevant trials.” As per point 8 of my feedback, you state the search will commence in November 2024. Please update the search date to include an end date.

Response: This has been clarified.

Comment: Page 7, line 118, “A copy of the Ovid MEDLINE search can be viewed in Supplementary Materials.” Please provide copies of electronic search strategies for all databases in the materials.

Response: As per the PRISMA-P guidelines only one search strategy is required. We have now added the others.

Comment: Page 7, lines 122-6, “We do recognise that the use of these terms has evolved over time and therefore, to make sure we do not exclude any relevant trials in MLTCs, we will screen the reference lists for selected studies and the previous reviews conducted in this area will also be screened for other potentially relevant publications (16, 22-24, 27). I suggest revising this sentence so it's shorter or split into 2 sentences.

Do you mean you will complete backward and forward citation screening?

Suggest text like this: Backward and forward citation tracking of included articles, that is, checking reference lists and searching for article citations, respectively, will be conducted in Web of Science and Google Scholar for additional literature unidentified by the search (minimising selection bias).

Response: We will not be conducted forward citation searching. We have amended the text to reflect that we will be conducting backwards citation searching and broken down the sentences.

Comment: Page 8, line 126, “We will also search the comprehensive database of the International Research Community in Multimorbidity (28).” Please add this database to the methods in the abstract.

Response: We have added this as suggested.

Comment: Page 8, line 141 “patient reported outcomes (e.g. HRQoL and measurement of physical or mental functioning) (30).” Are you including quality-of-life patient-reported outcomes? If so, please add.

Response: This has been amended to include QoL.

Comment: Page 8, line 143, “We will only include studies published as full text articles in English.” Why are you only limiting to English papers? You could translate in Google translate.

Response: We are not comfortable with using google translate in terms of the validity of the translations for complex medical terminology and potential bias. We will report how many studies are excluded due to this decision.

Comment: Page 9, line 156, study selection “All eligible studies from the literature search will be imported into Covidence (app.covidence.org/) where duplicates will be removed.” Please add a reference for Covidence.

Response: Reference added.

Comment: Page 9, line 197, onwards, I would separate out quality assessment and have this as its own section before you talk about data extraction as this is its own process. Regarding the quality assessment tools, I think you should specify the tools to be used, perhaps reference the EQUATOR network and provide examples Experimental studies | Study Designs | EQUATOR Network

Response: We have not carried out a quality assessment beyond assessing the risk of bias. We have amended this section to reflect this.

Comment: Page 9, line 197, “The risk of bias of the included studies will be assessed using the Cochrane Risk of Bias (RoB) tool (31). This tool assesses bias across five domains, and overall, each is rated as either low, high or some concerns in terms of the risk of bias.”

I would list the Cochrane ROB five domains it may orientate and inform the reader.

Response: We have added these.

Comment: How will the quality assessment be completed? i.e. independently by two reviewers?

Response: We are not completing a quality assessment, all data extracted including the risk of bias will be extracted independently by two reviewers. We have made this clearer in the text.

Comment: How will the quality assessments be presented? i.e. Tabulated in the summary of findings table

Response: We are not completing a quality assessment, we have made the synthesis section clearer in terms of how the risk of bias and CONSORT data will be displayed.

Comment: Page 11, line 207 onwards. Regarding data synthesis. Have you considered if and how the quality appraisal may affect the main results? Consider adding a statement to that effect.

Response: We have added a statement as suggested.

Comment: Page 12, line 228 onwards. Have you considered how else you will disseminate results other than peer-reviewed publications? For example, collaborators, advocacy organisations, social media etc.

Response: We will be sharing the results via webinars. The results will also be shared via social media and we have added this as suggested.

Discussion

Comment: Generally, I find the discussion section to be an unusual inclusion in a protocol paper. However, perhaps this is journal-centric. I would usually have a discussion section in my SR paper following the results. If this section remains, here are some comments for your consideration.

Response: We have followed the journal guidelines regarding manuscript sections.

Comment: Page 13, lines 235-7, states: “This comprehensive review of the methodological approach of trials conducted to inform the care of those living with MLTCs is in response to national and international calls for more research in this important growing clinical area.” Yet in the introduction, there appears to be more emphasis on national (perhaps UK-centric) issues. Please add a reference and align the focus across the paper.

Response: We have addressed the introduction regarding global interest and expanded this section to specifically cover this.

Comment: Page 13, lines 240-245 “Trial advancement was also a feature of the National Institute of Health and Care Research (NIHR) strategic framework for MLTCs, with methodologies being identified as one of the strategic priority areas. The NIHR has specifically noted the need for outcome frameworks and ensuring representation within MLTC trials. This review will benchmark current methodological approaches in MLTC trials, highlighting opportunities for improvement.

Please reference these statements. Is this only NIHR centric or have other organisations highlighted these issues?

Response: We have referenced these statements and updated this section to include the international context.

Comment: Page 13, lines 246-258. The text below would be better suited in the introduction as it justifies the methods, rationale, and focus of the review.

“Although previous reviews have been conducted in this area, they have been focussed on intervention effectiveness both generically (16) and focussed on specific types of intervention (such as models of care (33), behaviour change interventions (24), medication 249 adherence (34)), or on specific methodological issues such as ethnic representation (22) or MLTC subgroups, such as those with frailty plus MLTC (27). Unlike these previous studies we will not include terms related to comorbidity in our searches. This was a pragmatic decision to limit the number of studies needed to be screened and given that our interest is in trials of interventions aimed at those with MLTCs in the broadest sense and not those living with an index condition plus another (i.e., comorbidity). Any studies which have used terms related to comorbidity to describe a MLTC population will be picked up through screening the references of previous systematic reviews in this area. Given the terms multimorbidity and MLTC are becoming the more standard way over time to describe this patient group, we hope this will have a limited impact on more recent studies.”

Response: We thank the reviewer for this comment, we have moved the first part of this section to the introduction.

---

## [Decision Letter · Decision Letter 1]

25 Mar 2025

PONE-D-24-47106R1Randomised controlled trials for improving health outcomes for people living with multiple long-term conditions: protocol for a systematic review of methodological approaches, risk of bias and reporting qualityPLOS ONE

Dear Dr. Gray,

Thank you for submitting your manuscript to PLOS ONE. After careful consideration, we feel that it has merit but does not fully meet PLOS ONE’s publication criteria as it currently stands. Therefore, we invite you to submit a revised version of the manuscript that addresses the points raised during the review process.

We look forward to receiving your revised manuscript.

Kind regards,

Paolo Landa, Ph.D.

Academic Editor

PLOS ONE

Additional Editor Comments:

Please consider all the information provided by the reviewers in the revision process

Reviewers' comments:

Reviewer's Responses to Questions

**Comments to the Author**

1. Does the manuscript provide a valid rationale for the proposed study, with clearly identified and justified research questions?

Reviewer #2: Yes

Reviewer #3: Yes

Reviewer #4: Yes

Reviewer #5: Yes

2. Is the protocol technically sound and planned in a manner that will lead to a meaningful outcome and allow testing the stated hypotheses?

Reviewer #2: Yes

Reviewer #3: Yes

Reviewer #4: Yes

Reviewer #5: Yes

3. Is the methodology feasible and described in sufficient detail to allow the work to be replicable?

Reviewer #2: No

Reviewer #3: Yes

Reviewer #4: No

Reviewer #5: Yes

4. Have the authors described where all data underlying the findings will be made available when the study is complete?

Reviewer #2: No

Reviewer #3: Yes

Reviewer #4: Yes

Reviewer #5: Yes

5. Is the manuscript presented in an intelligible fashion and written in standard English?

Reviewer #2: Yes

Reviewer #3: Yes

Reviewer #4: No

Reviewer #5: Yes

6. Review Comments to the Author

You may also provide optional suggestions and comments to authors that they might find helpful in planning their study.

Reviewer #2: Thank you for pursuing research into this important topic.

Given your interest in trials specifically, I would assume you are searching Cochrane Central rather than the entire Cochrane Library - please clarify the abstract. I also find it odd to identify the interface in the abstract for only one of the databases; would suggest deferring that detail to the text.

In defining the topic, you use the terminology "long duration" - what does this mean specifically? How long?

Is the "two-thirds of adults aged over 65 will be affected" estimate specific to England, as in the preceding clause, or globally? Suggest clarifying.

"Despite more people living with MLTCs than single conditions" - what is the source for this claim? I'm not seeing it in the following reference.

The manuscript would benefit from a thorough copy-editing for clarity and flow.

The body of the manuscript should identify specifically the interfaces through which each database is being searched, and see comment above about Cochrane Library.

"The start date of January 1999 was chosen, as this is when the first trial in MLTCs was reported in a previous systematic review". The review referenced was limited to those taking place in primary care and community settings, which does not appear to be the case for your review - there may well be earlier trials in other contexts.

A rationale should be provided for limiting results to English.

Was consideration given to searching the grey literature? Its absence should be noted as a limitation.

What is the rationale for excluding secondary/subgroup analyses?

The Scopus search strategy appears to be missing parentheses around the multimorbidity concept. Also why not use the same proximity search strings as for the other databases?

The Medline search string presented appears to be missing several lines - the final line attempts to limit line 21, but the strategy before that point only has 17 lines.

It appears that some literature uses the alternative term "polymorbidity" - this should be included in the search.

Reviewer #3: Dear Authors,

It was interesting reading your revised protocol. The addressed topic is fundamental to improve the quality of life of people living with multiple long-term conditions. Your revised manuscript addressed almost all the previous reviewer comments which align completely with my feedback. A few points need to be re-addressed as follows:

1. It is acceptable to limit the inclusion of studies to English. However, it would be recommended to justify this language restriction as long as you are not searching for studies conducted only in the UK.

2. You did not separate the risk of bias into a different subsection, which is recommended for better structure and readability.

3. Lastly, it is recommended to fill out the PRISMA-P checklist and add it as a supplementary file.

Otherwise, your protocol is to be accepted and look forward toward reading your final results.

Reviewer #4: The analysis of methodological aspects of RCTs on multimorbidity is a valuable research aim. However, the language and structure of this protocol require significant improvements.

Major Issue = Inclusion/Exclusion Criteria

1) INDEX condition

The sentence: "In line with the reviews by Smith et al, trials with an index condition ... will be excluded." needs further clarification.

Specifically, if a trial includes patients with diabetes (DM) and hypertension (HTN), how will it be classified?

Does inclusion depend on how the study defines the patient population? For example:

If a study states, “we evaluated diabetic patients with hypertension,” does this make diabetes the index condition, leading to exclusion?

If it states, “we evaluated hypertensive patients with diabetes,” does this make hypertension the index condition, leading to exclusion?

If it simply states “we evaluated patients with diabetes and hypertension,” does this qualify as a multimorbidity study?!

or you would exclude studies on any specific combination of disorders and only include those RCT with multimorbidity? this part of selection is not clear.

These criteria must be explicitly defined so that others can replicate the systematic review and consistently determine which trials are included.

Minor issues:

1) Search Start Date (January 1999)

The authors justify using January 1999 as the starting point, stating that “this is when the first trial in MLTCs was reported in a previous systematic review.” However, the following reference suggests relevant trials exist before 1999: https://doi.org/10.7326/0003-4819-126-12-199706150-00004.

Given this, it may be more appropriate not to limit the search by a starting year unless a strong rationale is provided.

2) Grammatical and Stylistic Issues

The manuscript contains several grammatical errors. For instance, in the first two sentences of the introduction, the authors inconsistently use “is” and “are” when referring to MLTCs.

Careful proofreading is required to ensure grammatical consistency and readability.

3) Unprofessional Referencing

If the first sentence is a direct quotation, the source must be explicitly stated.

Additionally, citing four references for a simple statement is excessive and unprofessional. The authors should ensure that only the most relevant sources are cited.

Reviewer #5: The edited manuscript has been checked over as well as the comments and responses. Seems fine after responding to all the requested comments from the reviewers.

7. PLOS authors have the option to publish the peer review history of their article (what does this mean? ). If published, this will include your full peer review and any attached files.

**Do you want your identity to be public for this peer review?** For information about this choice, including consent withdrawal, please see our Privacy Policy .

Reviewer #2: No

Reviewer #3: No

Reviewer #4: No

Reviewer #5: No

---

## [Author Response · Author response to Decision Letter 2]

31 Mar 2025

Response to reviewers’ comments

Reviewer #2

Comment: Thank you for pursuing research into this important topic.

Response: We thank the reviewer for this positive comment.

Comment: Given your interest in trials specifically, I would assume you are searching Cochrane Central rather than the entire Cochrane Library - please clarify the abstract. I also find it odd to identify the interface in the abstract for only one of the databases; would suggest deferring that detail to the text.

Response: We have corrected and made consistent the list of databases used.

Comment: In defining the topic, you use the terminology "long duration" - what does this mean specifically? How long?

Response: We have added a definition of long-duration to the introduction.

Comment: Is the "two-thirds of adults aged over 65 will be affected" estimate specific to England, as in the preceding clause, or globally? Suggest clarifying.

Response: We have clarified that this estimate is specific to England.

Comment: "Despite more people living with MLTCs than single conditions" - what is the source for this claim? I'm not seeing it in the following reference.

Response: We apologise for this mistake, we have corrected the text.

Comment: The manuscript would benefit from a thorough copy-editing for clarity and flow.

Response: We have proof read the article and made improvements to the grammar and readability.

Comment: The body of the manuscript should identify specifically the interfaces through which each database is being searched, and see comment above about Cochrane Library.

Response: We have now identified both the databases and the interfaces.

Comment: "The start date of January 1999 was chosen, as this is when the first trial in MLTCs was reported in a previous systematic review". The review referenced was limited to those taking place in primary care and community settings, which does not appear to be the case for your review - there may well be earlier trials in other contexts.

Response:

Comment: A rationale should be provided for limiting results to English.

Response: We have now included the rationale.

Comment: Was consideration given to searching the grey literature? Its absence should be noted as a limitation.

Response: We have included this as a limitation in the discussion.

Comment: What is the rationale for excluding secondary/subgroup analyses?

Response: The rational has been added.

Comment: The Scopus search strategy appears to be missing parentheses around the multimorbidity concept. Also why not use the same proximity search strings as for the other databases?

Response: The search strategy has been checked by our librarian. The proximity settings for this database were changed to limit the size of the results, this change gave a reasonable number of results but was still able to identify the core papers included in other reviews.

Comment: The Medline search string presented appears to be missing several lines - the final line attempts to limit line 21, but the strategy before that point only has 17 lines.

Response: The medline search has been corrected.

Comment: It appears that some literature uses the alternative term "polymorbidity" - this should be included in the search.

Response: We thank the reviewer for this suggestion, we have now included this term in the search strategies.

Reviewer #3:

Comment: It was interesting reading your revised protocol. The addressed topic is fundamental to improve the quality of life of people living with multiple long-term conditions. Your revised manuscript addressed almost all the previous reviewer comments which align completely with my feedback.

Response: We thank the reviewer for their positive comment.

Comment: It is acceptable to limit the inclusion of studies to English. However, it would be recommended to justify this language restriction as long as you are not searching for studies conducted only in the UK.

Response: We are not limiting inclusion by the country the study was conducted in, this has been added.

Comment: You did not separate the risk of bias into a different subsection, which is recommended for better structure and readability.

Response: We have now separated the risk of bias into a subsection.

Comment: Lastly, it is recommended to fill out the PRISMA-P checklist and add it as a supplementary

file.

Response: We have completed a PRISMA-P checklist.

Comment: Otherwise, your protocol is to be accepted and look forward toward reading your final results.

Response: We thank the reviewer for their support.

Reviewer #4

Comment: The analysis of methodological aspects of RCTs on multimorbidity is a valuable research aim. However, the language and structure of this protocol require significant improvements.

Response: We thank the reviewer for their positive comment and have worked to improve the language and structure.

Comment: The sentence: "In line with the reviews by Smith et al, trials with an index condition ... will be excluded." needs further clarification.

Specifically, if a trial includes patients with diabetes (DM) and hypertension (HTN), how will it be classified?

Does inclusion depend on how the study defines the patient population? For example:

If a study states, “we evaluated diabetic patients with hypertension,” does this make diabetes the index condition, leading to exclusion?

If it states, “we evaluated hypertensive patients with diabetes,” does this make hypertension the index condition, leading to exclusion?

If it simply states “we evaluated patients with diabetes and hypertension,” does this qualify as a multimorbidity study?!

or you would exclude studies on any specific combination of disorders and only include those RCT with multimorbidity? this part of selection is not clear.

These criteria must be explicitly defined so that others can replicate the systematic review and consistently determine which trials are included.

Response: We have clarified the methods and added that trials of comorbidity will be excluded (i.e. studies which focus on 2 conditions specifically).

Comment: The authors justify using January 1999 as the starting point, stating that “this is when the first trial in MLTCs was reported in a previous systematic review.” However, the following reference suggests relevant trials exist before 1999: https://doi.org/10.7326/0003-4819-126-12-199706150-00004.

Given this, it may be more appropriate not to limit the search by a starting year unless a strong rationale is provided.

Response: The example study does not meet the inclusion criteria for the review, as diabetes is an index condition with hypertension as a comorbidity. We want the review to reflect current practice in terms of the design and analysis of trials for MLTC and therefore we believe this start date represents a pragmatic choice and has been used by other studies in this area.

Comment: The manuscript contains several grammatical errors. For instance, in the first two sentences of the introduction, the authors inconsistently use “is” and “are” when referring to MLTCs.

Careful proofreading is required to ensure grammatical consistency and readability.

Response: We have proof read the article and made improvements to the grammar and readability.

Comment: If the first sentence is a direct quotation, the source must be explicitly stated.

Additionally, citing four references for a simple statement is excessive and unprofessional. The authors should ensure that only the most relevant sources are cited.

Response: Thank you for your comment, given the definition is repeated in the methods we have removed the direct quotation from the introduction.

Reviewer #5

Comment: The edited manuscript has been checked over as well as the comments and responses. Seems fine after responding to all the requested comments from the reviewers.

Response: Thank you for your positive comment.

---

## [Decision Letter · Decision Letter 2]

24 Apr 2025

PONE-D-24-47106R2Randomised controlled trials for improving health outcomes for people living with multiple long-term conditions: protocol for a systematic review of methodological approaches, risk of bias and reporting qualityPLOS ONE

Dear Dr. Gray,

Thank you for submitting your manuscript to PLOS ONE. After careful consideration, we feel that it has merit but does not fully meet PLOS ONE’s publication criteria as it currently stands. Therefore, we invite you to submit a revised version of the manuscript that addresses the points raised during the review process.

**ACADEMIC EDITOR: the reviewers proposed a set of changes required in order to improve the manuscript. These modifications are essential in order to advance with the revision process. Please follow the indications provided by the reviewers.**

We look forward to receiving your revised manuscript.

Kind regards,

Paolo Landa, Ph.D.

Academic Editor

PLOS ONE

Reviewers' comments:

Reviewer's Responses to Questions

**Comments to the Author**

1. Does the manuscript provide a valid rationale for the proposed study, with clearly identified and justified research questions?

Reviewer #4: Yes

Reviewer #6: Partly

Reviewer #7: Yes

Reviewer #8: Yes

2. Is the protocol technically sound and planned in a manner that will lead to a meaningful outcome and allow testing the stated hypotheses?

Reviewer #4: Yes

Reviewer #6: Yes

Reviewer #7: Yes

Reviewer #8: Yes

3. Is the methodology feasible and described in sufficient detail to allow the work to be replicable?

Reviewer #4: Yes

Reviewer #6: Yes

Reviewer #7: Yes

Reviewer #8: Yes

4. Have the authors described where all data underlying the findings will be made available when the study is complete?

Reviewer #4: Yes

Reviewer #6: Yes

Reviewer #7: Yes

Reviewer #8: Yes

5. Is the manuscript presented in an intelligible fashion and written in standard English?

Reviewer #4: Yes

Reviewer #6: No

Reviewer #7: Yes

Reviewer #8: Yes

6. Review Comments to the Author

You may also provide optional suggestions and comments to authors that they might find helpful in planning their study.

Reviewer #4: Thank you for making these revisions. The quality of the manuscript has improved significantly. I have no further comments.

Reviewer #6: The manuscript improved greatly after the authors followed most of the reviewers comments.

I would like authors to reconsider some points regarding the reference section. some references have a link to download a file but when we click on it, it leads to empty web page.

Therefore, if the reference is a website, please input the link of the website instead of a link to a file download. also mention the date you access it to retrieve the information. (accessed on xx day xx month 202x year)

this reference also need to be rewritten:

International research community in multimorbidity. Multimorbidity publications 2021 [Available from: https://crmcspl-blog.recherche.usherbrooke.ca/.

This is a blog which has its own criteria for referencing:

Author of blog post. Year that the post was published or last updated. Title of blog post. Day/month of published or updated blog post. Available at: URL. (Accessed: date)

Do you mean all articles in this website page are this single reference?

Reviewer #7: Thank you for a sophisticated protocol for a systematic review. I have a few questions about your protocol.

"Not doing a meta-analysis due to the research question" needs some more clarification. You are including RCT studies and your PICO is in quantitative format. This means the results should be able to be running a meta-analysis. The reason does not appear to be valid.

Reviewer #8: I retaliate what the second and fourth reviewers said, that a thorough grammatical check should be carried out before submitting the manuscript. (Check out line 197 and 198 and rectify them to make grammatical sense).

Otherwise thank you for choosing to concentrate on this unique research area.

7. PLOS authors have the option to publish the peer review history of their article (what does this mean? ). If published, this will include your full peer review and any attached files.

**Do you want your identity to be public for this peer review?** For information about this choice, including consent withdrawal, please see our Privacy Policy .

Reviewer #4: No

Reviewer #6: **Yes: ** Doaa Husham Majeed Al-saadi

Reviewer #7: **Yes: ** Thomas Aaron Ricks

Reviewer #8: No

---

## [Author Response · Author response to Decision Letter 3]

15 May 2025

Response to reviewers

Comment: I would like authors to reconsider some points regarding the reference section. some references have a link to download a file but when we click on it, it leads to empty web page.

Response: We have updated the references, paying specific attention to those of websites/blogs.

Comment: Do you mean all articles in this website page are this single reference?

Response: Yes, we have updated the text to make this clearer.

Comment: “Not doing a meta-analysis due to the research question" needs some more clarification. You are including RCT studies and your PICO is in quantitative format. This means the results should be able to be running a meta-analysis. The reason does not appear to be valid.

Response: The research question for this review does not consider effectiveness and therefore conducting a meta-analysis is not required to answer the research question. We are interested in the current approaches to design and analysis for MLTC trials. The results from this study will feed into a Delphi study which will prioritise research uncertainties in this area. This type of systematic review is not uncommon in methodological research and can be highly cited as useful for driving future research in this area. A few examples are given below

Eldridge SM, Ashby D, Feder GS, Rudnicka AR, Ukoumunne OC. Lessons for cluster randomized trials in the twenty-first century: a systematic review of trials in primary care. Clin Trials. 2004 Feb;1(1):80-90. doi: 10.1191/1740774504cn006rr. PMID: 16281464.

Varghese E, Briola A, Kennel T, Pooley A, Parker RA. A systematic review of stepped wedge cluster randomized trials in high impact journals: assessing the design, rationale, and analysis. J Clin Epidemiol. 2025 Feb;178:111622. doi: 10.1016/j.jclinepi.2024.111622. Epub 2024 Dec 2. PMID: 39631553.

Comment: I retaliate what the second and fourth reviewers said, that a thorough grammatical check should be carried out before submitting the manuscript. (Check out line 197 and 198 and rectify them to make grammatical sense).

Response: We have completed a thorough review of the grammar, paying particular attention to the lines suggested by the reviewer.

---

## [Editor Report · Decision Letter 3]

18 May 2025

Randomised controlled trials for improving health outcomes for people living with multiple long-term conditions: protocol for a systematic review of methodological approaches, risk of bias and reporting quality

PONE-D-24-47106R3

Dear Dr. Gray,

We’re pleased to inform you that your manuscript has been judged scientifically suitable for publication and will be formally accepted for publication once it meets all outstanding technical requirements.

Kind regards,

Paolo Landa, Ph.D.

Academic Editor

PLOS ONE
---

## [Editor Report · Acceptance letter]

PONE-D-24-47106R3

PLOS ONE

Dear Dr. Gray,

I'm pleased to inform you that your manuscript has been deemed suitable for publication in PLOS ONE. Congratulations! Your manuscript is now being handed over to our production team.

Kind regards,

on behalf of

Dr. Paolo Landa

Academic Editor

PLOS ONE